# Selecting Key Views for Zero-Shot Entity Linking

**Xuhui Sui**[1], **Ying Zhang**[1,*], **Kehui Song**[1],
**Baohang Zhou**[1], **Xiaojie Yuan**[1], **Wensheng Zhang**[2]

[1] College of Computer Science, VCIP, TMCC, TBI Center, Nankai University, China
[2] Institute of Automation, Chinese Academy of Sciences, China
{suixuhui,songkehui,zhoubaohang}@dbis.nankai.edu.cn
{yingzhang,yuanxj}@nankai.edu.cn, zhangwensheng@hotmail.com

## Abstract

Entity linking, which aligns mentions in the text to entities in knowledge bases, is essential for many natural language processing tasks. Considering the real-world scenarios, recent research hotspot of entity linking has focused on the zero-shot setting, where mentions need to link to unseen entities and only the description of each entity is provided. This task challenges the language understanding ability of models to capture the coherence evidence between the mention context and entity description. However, entity descriptions often contain rich information from multiple views, and a mention with context only relates to a small part of the information. Other irrelevant information will introduce noise, which interferes with models to make the right judgments. Furthermore, the existence of these information also makes it difficult to synthesize key information. To solve these problems, we select key views from descriptions and propose a KVZEL framework for zero-shot entity linking. Specifically, our KVZEL first adopts unsupervised clustering to form sub views. Then, it employs a mention-aware key views selection module to iteratively accumulate mention-focused views. This puts emphasis on capturing mention-related information and allows long-range key information integration. Finally, we aggregate key views to make the final decision. Experimental results show the effectiveness of our KVZEL and it achieves the new state-of-the-art on the zero-shot entity linking dataset.

## 1 Introduction

Entity linking (EL) is the task of assigning ambiguous mentions in textual input to their corresponding entities in a given real-world knowledge base. EL, as a fundamental task in the information extraction area, plays an important role in many downstream natural language processing (NLP) applications,

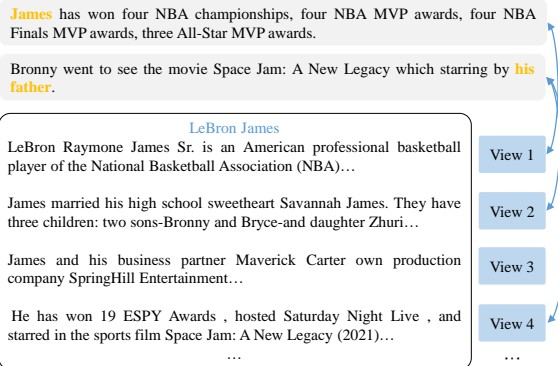

(a) An example of entity description with many views.

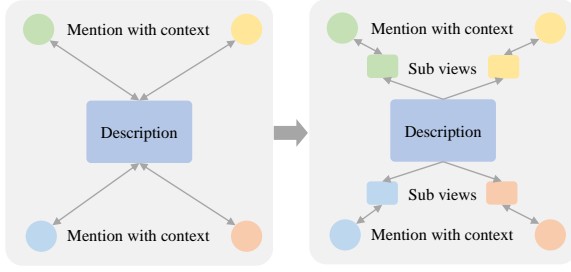

(b) Other methods and our proposed framework.

Figure 1: The motivation illustration of our proposed framework. Usually, an entity can be linked by different mentions from multiple views. Our proposed framework comes from this observation and selects key views from entity descriptions to better align mentions with divergent context.

such as semantic search (Blanco et al., 2015), content analysis (Huang et al., 2018) and question answering (Welbl et al., 2018; Li et al., 2020). Traditional EL approaches are built on the assumption that the train and test set share the same entity distributions, which means that linked entities in the test set are seen during training. However, in many real-world scenarios, the process of labeling data is labor-intensive and error-prone. More importantly, for some specialized domains such as medical and legal domains, the training data is sensitive or proprietary. Thus, it is not easy to obtain labeled data for these domains. Therefore, EL models need to

---
*Corresponding author.

have the capability of generalizing to new entities of new domains.

To address this, the zero-shot entity linking task (Logeswaran et al., 2019) has been proposed, where mentions need to be linked to unseen entities of new domains. Moreover, considering that there is no external knowledge (e.g. frequency statistics and structured data) in many specialized domains, the textual description for each entity is the only information provided in this task. Based on the above, we can see that the entity description, as the only identification information of entities, is crucial for this task. Some previous works (Yao et al., 2020; Tang et al., 2021) have realized its importance and constructed models to circumvent the length limitation problem of BERT-based models to capture richer entity description information.

However, to the best of our knowledge, all previous methods utilize the same entire entity description to match different mentions with divergent context. This seems struggling to manage entities with rich multiple views information. We argue that only part of description information is related to mentions and other irrelevant information will introduce noise and make models reduce emphasis on the corresponding details. An example shown in Figure 1a, the entity description of "LeBron James" contains multiple views information, such as his NBA journey, his family, etc. For the first mention, it only relates to "View 1" in the entity description. Other views are useless in determining whether the "James" should link to the entity "LeBron James". For the second mention, it is related to "View 2" and "View 4". However, the description of "LeBron James" mainly concentrates on his professional journey, which will introduce noise to interfere with entity linking models matching "his father" and "LeBron James". Furthermore, "View 2" and "View 4" may exist in different parts of the description, which makes it not trivial to synthesize the key information of the two views.

To address these problems, in this paper, we try to select key views from entity descriptions for different mentions with divergent context, which is shown in Figure 1b. This can also help entity linking models solve the problem of length limitation in BERT-based models, since the selected views are only a part of entity description. We propose a novel **K**ey **V**iews selection framework for **Z**ero-shot **E**ntity **L**inking in this paper, **KVZEL** in short. KVZEL first adopts unsupervised clustering

to group the sentences, and those in each cluster compose a sub view. When judging whether a mention in text matches the entity, humans tend to read the whole description and only focus on the view that is most relevant to the mention. Considering that this mention may be related to multiple views of entity description, after finding that a single view is insufficient to make the judgment, humans will use the previously relevant views as clues and move on to find the next relevant view. With the inspiration of the human's thinking process, our KVZEL designs a mention-aware key views selection module to iteratively accumulate the most relevant views. These views are then aggregated into a new text, which serves as the mention-focused and highly-condensed version of original entity description. Finally, the text replaces the description and is utilized by our entity linking module to make the final decision.

To summarize, our major contributions are shown as follows:

- To the best of our knowledge, this work is the first to select key views from entity descriptions to link different mentions with divergent context. This is very meaningful for entities with rich multiple views information.

- We propose a novel framework KVZEL for zero-shot entity linking, which imitates the human's thinking process to iteratively accumulate mention-related views. These views are aggregated to measure the matching score between mentions and entities.

- Experimental results on zero-shot entity linking dataset (Logeswaran et al., 2019) show that our KVZEL achieves new state-of-the-art performance. Further analysis demonstrates the effectiveness of our framework.

## 2 Related Work

### 2.1 Entity Linking

Entity linking, as a bridge between real-world text and standard knowledge base, has been widely explored in recent years. Many entity linking methods (Sun et al., 2015; Yamada et al., 2016; Le and Titov, 2018; Yang et al., 2019; Tedeschi et al., 2021; Barba et al., 2022) have achieved great success in both academic research and industrial applications. While these results are impressive, the models in these methods are learned under the setting where

Figure 2: The overall architecture of our KVZEL framework, which includes three modules: a sub view clustering module, a mention-aware key views selection module and an entity linking module.

linked entities in the test set are available during training. Furthermore, with the expansion of knowledge in knowledge bases, many state-of-the-art methods (Ganea and Hofmann, 2017; Kundu et al., 2018; Raiman and Raiman, 2018; Orr et al., 2021) have made full use of external knowledge. However, in many real-world scenarios, labeled data and external knowledge are not easily available in many specialized domains. Based on empirical evidence, the performance of models deteriorates substantially when dealing with new entities of new domains without external knowledge. This motivates many researchers to study zero-shot entity linking (Logeswaran et al., 2019).

## 2.2 Zero-Shot Entity Linking

Zero-shot entity linking is the task where mentions must be linked to unseen entities and only the description of each entity is available. It consists of two phases: candidate generation (Wu et al., 2020; Sun et al., 2022; Sui et al., 2022) and candidate ranking (Yao et al., 2020; Tang et al., 2021). Following previous works (Logeswaran et al., 2019; Yao et al., 2020; Tang et al., 2021), we only focus on the candidate ranking phase and use the traditional Information Retrieval (IR) approach of BM25 (Robertson et al., 1994) in the candidate generation phase. Logeswaran et al. (2019) uses a cross-encoder architecture that concatenates mention context and entity description and feeds it into BERT (Devlin et al., 2019) to produce the matching score by performing cross-attention. However, BERT-based models may lose important description information due to the limitation of their input

length. Yao et al. (2020) solves the problem by initializing larger position embeddings to expand the effective reading length of BERT. Tang et al. (2021) proposes a bidirectional multi-paragraph reading model which segments a long description into short paragraphs to make use of more description information. However, the consideration of paragraphs that are irrelevant to the mention may introduce noise in the final representation and limit the long-range dependencies between relevant information. This motivates us to select key views from entity descriptions for different mentions with divergent context to address these problems.

## 3 Methodology

Figure 2 shows the overall architecture of our proposed KVZEL framework. We first use the sub view clustering module to generate sub views from descriptions. Then we design a mention-aware key views selection module to select relevant sub views for different mentions with divergent context. Finally, our entity linking module makes the final decision based on the selected views. In this section, we introduce these three modules in detail.

## 3.1 Sub View Clustering

The goal of our sub view clustering module is to cluster sentence chunks describing the same view together, so as to select key views later. For each given entity description, we first segment it into sentence chunks $S = [s_1, s_2, ..., s_L]$. Intuitively, each individual sentence in the description is taken as a chunk. However, the understanding of short sentences is often not easy and needs the help of

context. A simple solution is to segment the entire description into chunks of fixed length. However, this destroys the syntactic relationship and may lead to clustering errors. Thus, we fuse these two ways and set a fixed capacity $c$ for each chunk. Each chunk loads as many sentences as possible. After loading sentences, the sentence sequences in chunks are padded to $c$ with zero values.

We feed the sentence chunks $S$ into BERT (Devlin et al., 2019) in the form of [CLS] $s_i$ [SEP] to extract features, which are the vectors in the last hidden layer corresponding to the position of [CLS] tokens. Then, we adopt mean-shift (Cheng, 1995) clustering approach to group sentence chunks. Compared to other traditional unsupervised clustering approaches (such as k-means (Lloyd, 1982), spectral clustering (Ng et al., 2001), etc), mean-shift does not need to specify the number of clusters, which is more suitable for our application scenario. Those sentence chunks in each cluster compose a sub view:

$$V = \text{Mean-Shift}(\text{BERT}(S; \theta_{\text{BERT}_1}))$$

## 3.2 Mention-Aware Key Views Selection

The goal of our mention-aware key views selection module is to iteratively accumulate mention-related views for the final entity linking process. This is similar to human reading through the whole description view by view to measure the relevance of the mention and each view. Then they will focus on the view that is most relevant to the mention. When moving on to find the next relevant view, they will still remember previously relevant views and utilize them as clues to help this process.

### 3.2.1 Feature Extraction

For a given mention $m$, suppose there are $P$ candidate entities of $m$ and we have $n$ sub views $V = [V_1, V_2, ..., V_n]$ of the $p$-th candidate entity $e_p$. In the initial stage, our clues only consist of the mention $m$ and the entity title:

$$[\text{CLS}] \, me \, [\text{SEP}] \, t \quad (1)$$

where $me$ and $t$ are the inputs of the mention $m$ and the entity title. Following Wu et al. (2020), $me$ and $t$ are constructed as: $me = cl$ [Ms] $mention$ [Me] $cr$ and $t = title$ [ENT], where $mention$, $cl$, $cr$, $title$ are the word-piece tokens of the mention, context before and after the mention and the entity title respectively. [Ms] and [Me] are special tokens to tag the mention, and

[ENT] is the special token to separate entity title and entity description. The clues are concatenated with the $n$ views separately and a special separator token [SEP] is added at the end. We feed them into a BERT encoder ($\theta_{\text{BERT}_2}$) to extract features. It produces a matrix representation $X = [x_1, x_2, ..., x_n]$ to represent the clues-view pairs, where $x_i \in \mathbb{R}^d$ is the output of the last hidden layer corresponding to the position of the [CLS] token for the $i$-th view and $d$ is the dimension of hidden states of BERT.

### 3.2.2 Module Training

Since the process of labeling data is labor-intensive and error-prone, we do not have any labels about key views. Therefore, we introduce a matching task to enable our module to learn relevance scores between mentions and views. We argue that views that are more relevant to the mention can contribute more to determining whether the mention and the candidate entity are matched.

Specifically, we use a trainable weight parameter $W_a \in \mathbb{R}^{1 \times d}$ as one of the inputs and utilize the SoftMax function to calculate the attention scores of the matching result with regard to the views:

$$G = \text{SoftMax}(W_a X^T)$$

where $G \in \mathbb{R}^{1 \times n}$. Then, we utilize the weighted aggregation with the attention scores to obtain the mention-entity representation: $Z = GX$, where $Z \in \mathbb{R}^d$. Finally, we get the binary prediction $\hat{y}$ by feeding the representation into a feed-forward neural network FFNN: $\hat{y} = \text{FFNN}(Z; \theta_{F_1})$. The prediction measures whether the mention and the candidate entity are matched.

We employ cross-entropy as our loss objective, which is calculated as follows:

$$\mathcal{L}_{Selection} = -(y\log\hat{y} + (1 - y)\log(1 - \hat{y})) \quad (2)$$

where $y$ takes the value 1 if the mention and the candidate entity are matched.

### 3.2.3 Key Views Selection

During the matching task training, the vector $G$ learns to measure the degree to which each view contributes to determining whether mentions and entities are matched. In the inference stage, we use it as the relevance scores between clues and all views of the candidate entity and utilize an Argmax function to obtain the most key view:

$$k = \text{Argmax}(G) \quad (3)$$

**Algorithm 1:** The key views selection process of our KVZEL.

**Input:** The mention with context $m$,
candidate entity $e_p$, candidate view
list $V$ of $e_p$, number of iterations $Q$.

**Output:** The selected view list $\hat{V}$.

1 Construct initial clues by Eq. 1.
2 **for** *each iteration $q = 1$ to $Q$* **do**
3    Calculate $\mathcal{L}_{Selection}$ by Eq. 2 and train our key views selection module.
4    Calculate the most key view $V_k$ by Eq. 3 in the inference stage.
5    Add $V_k$ to existing clues and the selected view list $\hat{V}$.
6    Remove $V_k$ from the candidate view list $V$.
7 **end**
8 **return** $\hat{V}$

| Domains | Length | Entities | Mentions |
|---|---|---|---|
| Training | | | |
| American Football | 665.06 | 31929 | 3898 |
| Doctor Who | 264.14 | 40281 | 8334 |
| Fallout | 229.68 | 16992 | 3286 |
| Final Fantasy | 497.35 | 14044 | 6041 |
| Military | 870.55 | 104520 | 13063 |
| Pro Wrestling | 639.53 | 10133 | 1392 |
| StarWars | 379.76 | 87056 | 11824 |
| World of Warcraft | 242.16 | 27677 | 1437 |
| Validation | | | |
| Coronation Street | 264.55 | 17809 | 1464 |
| Muppets | 161.50 | 21344 | 2028 |
| Ice Hockey | 282.62 | 28684 | 2233 |
| Elder Scrolls | 269.03 | 21712 | 4275 |
| Test | | | |
| Forgotten Realms | 257.29 | 15603 | 1200 |
| Lego | 223.86 | 10076 | 1199 |
| Star Trek | 393.21 | 34430 | 4227 |
| YuGiOh | 643.68 | 10031 | 3374 |

Table 1: Overall statistics of the Zeshel dataset, which consists of the average length of documents, the number of entities and labeled mentions.

where $k$ is the view marker. Then, we add this view $V_k$ (e.g. $V_2$) to our existing clues. And we will not consider the view $V_k$ in the next iteration. We summarize the key views selection process of our KVZEL in Algorithm 1.

### 3.3 Entity Linking

The goal of our entity linking module is to aggregate key views into a new text $\hat{des}$ to make the final decision. Considering that the first view sentences (view $V_0$) are able to generally describe the entity, we first add the first view $V_0$ to $\hat{des}$ to enhance the overall understanding of the entity. Then we sort the selected key views to maintain the relative ordering in the original description to preserve the syntactic dependencies. The key views are then added to $\hat{des}$ one by one. The new text $\hat{des}$ serves as the mention $m$ focused highly-condensed version of the entity description of $e_p$. Finally, $\hat{des}$ replaces the original entity description and is read by our entity linking module.

Following Wu et al. (2020) and similar to Eq. 1, the input text of the mention $m$ and the candidate entity $e_p$ is constructed as: [CLS] $me$ [SEP] $t\ \hat{des}$ [SEP]. Then the input text is entered into a BERT encoder ($\theta_{\text{BERT}_3}$) to produce the vector representation $U_p$, which is the output of the last hidden layer corresponding to the position of the [CLS] token. Finally, the representation $U_p$ is input into a feed-forward neural network FFNN and the score $s_p$ between the mention $m$ and its $p$-th candidate

entity $e_p$ is calculated using a SoftMax function:

$$\hat{s}_p = \text{FFNN}(U_p; \theta_{F_2}), \ s_p = \frac{\exp(\hat{s}_p)}{\sum_{j=1}^{P} \exp(\hat{s}_j)}$$

where P is the number of candidate entities of the mention $m$. We employ cross-entropy as our loss objective, which is calculated as follows:

$$\mathcal{L}_{Linking} = -\sum_{p=1}^{P} y_p \log s_p + (1 - y_p) \log(1 - s_p)$$

where $y_p \in \{0, 1\}$ and $y_p$ takes the value 1 if the candidate entity $e_p$ is the gold entity of the mention $m$, otherwise it takes the value 0.

## 4 Experiments

### 4.1 Dataset

Following recent zero-shot entity linking works (Logeswaran et al., 2019; Wu et al., 2020; Yao et al., 2020; Tang et al., 2021), we evaluate our framework KVZEL under the Zeshel dataset,[1] which is constructed by Logeswaran et al. (2019). Overall statistics of the dataset are shown in Table 1. The dataset was built based on the documents in Wikia.[2] Wikia consists of encyclopedias, each specialized

[1] https://github.com/lajanugen/zeshel
[2] https://www.wikia.com

| Methods | Forgotten Realms | Lego | Star Trek | YuGiOh | Macro Acc. | Micro Acc. |
|---|---|---|---|---|---|---|
| Cross-encoder | 85.60 | 76.90 | 75.80 | 67.22 | 76.38 | 74.21 |
| $E_{repeat}$ | – | – | – | – | 77.58 | – |
| Uni-MPR | 85.55 | 77.42 | 78.23 | 68.29 | 77.62 | 75.78 |
| Bi-MPR | **89.09** | 77.18 | 79.20 | 69.98 | 78.61 | 76.70 |
| KVZEL w/o iteration | 87.10 | 76.90 | 78.42 | 71.36 | 78.45 | 77.35 |
| KVZEL w/o $V_0$ | 88.30 | 76.08 | 78.96 | 73.41 | 79.19 | 78.24 |
| KVZEL (Ours) | 88.80 | **77.93** | **80.65** | **73.50** | **80.22** | **79.30** |

Table 2: Results on the test set of Zeshel dataset. Macro Acc. represents the average accuracy of these four test domains. Micro Acc. represents the weighted average accuracy of these four domains. In the results, the highest values are in bold and the results unavailable are left blank. All scores are averaged 5 runs using different random seeds, and our results over all baselines are statistically significant with $p < 0.05$ with the t-test.

in a particular subject such as sports and film series. Each particular subject can be considered as a domain. Zeshel consists of 16 specialized domains, 8 domains for training, 4 for validation, and 4 for test. Each domain has its individual entity dictionary, allowing the performance evaluation on entire unseen entities of new domains. The training set has 49,275 labeled mentions while the validation and test sets both have 10,000 unseen mentions.

## 4.2 Implementation Details

For a fair comparison, we use the BERT-base-uncased (Devlin et al., 2019) as our encoder and use the accuracy as our evaluation metric. For the sub view clustering module, the capacity $c$ is set to 64. For the mention-aware key views selection module, following Tang et al. (2021), we set the maximum sequence length of the mention context to be 256. We train this module 1 epoch per iteration using a batch size of 8 and a learning rate of 2e-5. The number of iterations $Q$ is set to 3, which can get the best results. We use the AdamW (Loshchilov and Hutter, 2017) optimizer to optimize our module. For the entity linking module, also following Tang et al. (2021), the max length for the mention context and entity description are both set to 256. We train this module in 5 epochs. All parameters are also optimized by AdamW, with learning rate 2e-5, and 10% warmup steps. Experiments were conducted on two NVIDIA GeForce RTX A6000 GPUs with 48 GB memory.

## 4.3 Comparison Methods

For the quantitative evaluation of our KVZEL, we use the following state-of-the-art methods (Logeswaran et al., 2019; Wu et al., 2020; Yao et al., 2020; Tang et al., 2021) for comparison. All of these methods including our KVZEL are based on

the cross-encoder (Logeswaran et al., 2019; Wu et al., 2020), which feeds the concatenation of mention context and entity description to BERT and outputs the [CLS] token's embeddings to produce the relevance scores. $E_{repeat}$ (Yao et al., 2020) extends the effective reading length of BERT by repeating the position embedding to solve the long-range modeling problem in entity descriptions. Uni-MPR (Tang et al., 2021) segments a long description into short paragraphs to make use of more description information. Bi-MPR (Tang et al., 2021) is based on Uni-MPR, which segments mention context into short paragraphs to incorporate more mention information. Note that we do not consider any domain-adaptive techniques in our KVZEL and comparison methods, which improves the final performance but outsides our work scope.

## 4.4 Overall Performance

The comparison results on the Zeshel dataset are shown in Table 2. We can observe that our KVZEL outperforms all baseline methods on average, illustrating the effectiveness of our KVZEL and the superiority of selecting key views for different mentions with divergent context. It's worth noting that all baselines perform worse on the YuGiOh domain. We conjecture that this is because the entities in this domain has more views since we find that the entities with longer descriptions contain more views. This is consistent with our claim that previous methods seem struggling to manage entities with rich multiple views information. We observe that our KVZEL makes a significant improvement on the domains (e.g. Star Trek and YuGiOh) whose descriptions have rich multiple views information. Our KVZEL avoids the noise brought by irrelevant view information and allows long-range key information integration, which is very effective for these

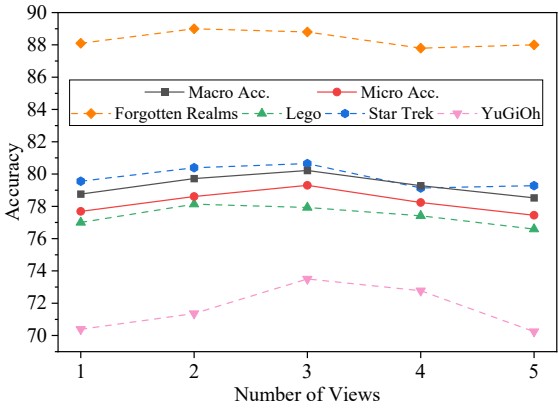

Figure 3: The effect of the number of selected views. Dashed lines indicate the accuracy on the four test domains and solid lines indicate the average accuracy.

domains. In general, our KVZEL outperforms the baseline methods for 1.61% and 2.60% on Macro Acc. and Micro Acc. on the test set of Zeshel dataset, respectively.

## 5 Analysis

### 5.1 Ablation Study

To better understand our KVZEL framework, we investigate the contribution of some components through ablation studies, and the results are also shown in Table 2. Considering that our entity linking module is based on cross-encoder, the comparison results to cross-encoder could also be regarded as the ablation study to justify the advantage of our KVZEL. To evaluate the contribution of iteratively accumulating key views, we train the key views selection module only once and select the same number of key views for the entity linking module. We find that this leads to a significant performance drop, which demonstrates the effectiveness of utilizing the previously selected views as clues to move on to find the next key view. We then do not add the first view $V_0$ to $\tilde{des}$ and only consider the selected views in our entity linking module. It can be observed that KVZEL w/o $V_0$ performs worse than KVZEL. This indicates that the adding of $V_0$ helps our entity linking module understand entities better, since we observe the first view sentences of most entities are able to generally describe the entity.

### 5.2 Effect of Number of Selected Views

We also explore the effect of different numbers of selected views on our KVZEL performance. The experimental results are shown in Figure 3. We

| Length | Number | Proportion |
|---|---|---|
| (0, 200) | 2752 | 40.40% |
| [200, 500) | 2025 | 29.73% |
| [500, 1000) | 954 | 14.00% |
| [1000, 2000) | 554 | 8.13% |
| [2000, +∞) | 527 | 7.74% |

Table 3: The statistics of entities with different description lengths. We partition the entities by the tokens number in entity descriptions.

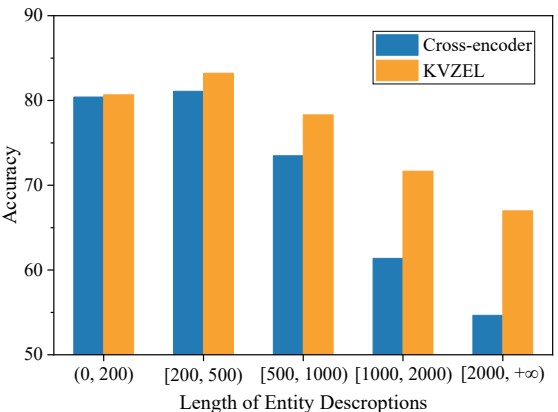

Figure 4: The effect of cross-encoder and our KVZEL on entities with different description lengths. We calculate metrics based on the partition in Table 3.

can find that the required number of selected views changes with different domains. For domains with rich views in entity descriptions (e.g. YuGiOh), the integration of more information from multiple views becomes particularly important. While for domains whose views are not as rich (e.g. Forgotten Realms, Lego), single or two views are sufficient to help our entity linking module to correctly link to gold entities. In general, we can observe that our KVZEL with 3 selected views performs best on average. If the number of selected views is too small, there will be not enough information to help our entity linking module make the correct decision. Conversely, if this number is too large, this will introduce noise from irrelevant views and cause the performance to drop.

### 5.3 Effect of Description Length

Through our observation, we find that the entities with longer descriptions contain more views. To further show the effectiveness of our proposed KVZEL framework, we evaluate the effect of cross-encoder and our KVZEL on entities with different description lengths. The statistics of entities on the test set of Zeshel dataset are shown in Table 3 and

| Mention with context | Description input of baselines | Description input of KVZEL |
|---|---|---|
| Lucifer reuses the **cat** single-stud mould introduced for the Friends theme. It features Lucifer standing up with bent back, elevated head looking straight ahead and long tail lying flat curled... | The Cat is a Friends animal figure introduced in 2013. The black one, released in the first series of Friends Animals, became available even in December 2012... | **The first selected view:** The cat is a single-stud mould, which features it standing up with bent back, elevated head looking straight ahead and long tail lying flat curled around the left side... |
| Having just defeated **Sherry**, Akiza walks over to her. Sherry protests that Akiza ran away from the Academy. Akiza admits that she ran away, and acknowledges that Duel Academy Sanctuary may foster elite... She asks Akiza where she learned the trick of riding her Duel Runner with her eyes shut... | Sherry LeBlanc is a Psychic Duelist with an ability referred to as Hand Scan, in which she can view her opponent's current hand. Design. Appearance. Sherry has long, waist length, blond hair with bangs that curl away from her face in a horizontal direction and has emerald color eyes... | **The first selected view:** Akiza chose to leave the academy due to it's elitist ways, which made Sherry feel betrayed and harbor a hatred for Akiza... **The second selected view:** After the Duel, Sherry and Akiza become friends again after Sherry questions Akiza on where she learned to drive a Duel Runner with her eyes closed... |

Table 4: Qualitative analysis of test samples in Zeshel dataset. We analyze the gold entity description input of baselines and our KVZEL. The mentions in text are written in bold.

the experimental results are shown in Figure 4. We can find that the cross-encoder (without selecting key views) obtains passable performance on entities with few views but fails to manage those multi-view entities as the length of descriptions increases. For example, the accuracy of cross-encoder is over 80% for entities with 0-500 description tokens but about 60% for entities with over 1000 description tokens. This is the limitation for zero-shot entity linking. Our KVZEL avoids the noise brought by irrelevant view information and allows long-range key information integration by iteratively accumulate key views. This helps our framework manage the entities with rich multiple views information. Our KVZEL significantly improves the accuracy to 71.66% (+10.29%) and 66.98% (+12.33%) for entities with 1000-2000 tokens and over 2000 tokens, respectively. Furthermore, considering its ability to avoid noise brought by irrelevant views information, our KVZEL also slightly improves the performance on entities with few views, which achieves the accuracy of 80.67% (+0.26%) and 83.21% (+2.12%) for entities with 0-200 tokens and 200-500 tokens respectively.

### 5.4 Case Study

To conduct a qualitative analysis, Table 4 shows two examples from the test set of Zeshel dataset. While the baseline methods tend to input the entire entity descriptions, our KVZEL only inputs the mention-focused views of entity descriptions. This indeed helps our framework avoid noise brought by irrelevant information and enhances emphasis on the corresponding details. For these two examples, our KVZEL can point to the gold entities "Cat (Friends)" and "Sherry LeBlanc (manga)" instead of "Catherine Cat" and "Sherry LeBlanc" like baselines. This indicates that baseline methods can only capture the rough meaning of a description from a general view, while our KVZEL is able to dive into the descriptions and capture more fine-grained semantic information from key views. Furthermore, for the second example, the mention with context is related to multiple views. A key observation is that our KVZEL can precisely select key views to aggregate to make the final decision, which indicates that our KVZEL has the ability of long-range key information integration.

## 6 Conclusion

In this paper, we explore an issue that was overlooked when only textual descriptions are available in zero-shot entity linking scenarios, that is, only part of description information is related to mentions and other irrelevant information will introduce noise and make models reduce emphasis on the corresponding details. To overcome this issue, we propose a KVZEL framework to select key views from entity descriptions for different mentions with divergent context. Specifically, it first adopts unsupervised clustering to form sub views. Then, it imitates the human's thinking process to iteratively accumulate mention-related views, which helps our entity linking module have the ability of long-range information integration. The selected views are aggregated into a new text, which replaces the original description as the input of our entity linking module to make the final decision. Experimental results demonstrate that the proposed KVZEL framework achieves new state-of-the-art performance on the zero-shot entity linking dataset.

## Limitations

Although our KVZEL achieves new state-of-the-art for zero-shot entity linking, it also poses several limitations as follows: 1) In this work, for simplicity, we only use the traditional unsupervised clustering approach mean-shift to form sub views. However, other state-of-the-art clustering approaches also can be used and may work better. We leave the selection of a more effective cluster approach to future work. 2) In this work, we only focus on the zero-shot entity linking setting, which is a low-resource scenario and only provides the textual description information for each entity. This task challenges the ability of entity linking models to understand the language consistency between mention contexts and entity descriptions. Our KVZEL improves this ability by selecting key views from entity descriptions. Therefore, when our KVZEL is extended to other entity linking settings which may have much external knowledge (e.g. frequency statistics and structure data, etc.) and not focus on the language understanding ability so much, the improvement of performance may be insignificant.

## Acknowledgments

This research is supported by the National Natural Science Foundation of China (No. 62272250, 62002178), the Natural Science Foundation of Tianjin, China (No. 22JCJQJC00150).

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
