# OpenReview forum: "Selecting Key Views for Zero-Shot Entity Linking"
_EMNLP/2023/Conference — EMNLP 2023 Findings_

### Official Review · Reviewer_zpya · 2023-08-02

**Soundness:** 3

**Excitement:**

3: Ambivalent: It has merits (e.g., it reports state-of-the-art results, the idea is nice), but there are key weaknesses (e.g., it describes incremental work), and it can significantly benefit from another round of revision. However, I won't object to accepting it if my co-reviewers champion it.

**Paper Topic And Main Contributions:**

The paper focuses on the problem of zero-shot entity linking where only entity description has been provided. The authors argue that only part of the provided description is related to the mention and rest of the part introduce noise and divert the attention of the model from the actual answer. For this, they propose KVZEL, Key View selection framework for Zero-shot Entity Linking. This involves breaking an entity description into multiple sentence chunks which are embedded using an BERT model and are clustered into multiple views using mean-shift clustering approach. For a given mention, a set of views is selected which is then utilized intead of the entity description. This helps to fit the relevant part of description within BERT model max sequence length.

**Reasons To Accept:**

Strengths:
1. The paper is well written and the proposed approach is interesting.
2. The propose method performs better than the baseline approaches on Zeshel dataset across multiple domains. The ablation results show that the first selected view plays a vital role on the performance of the model.


**Reasons To Reject:**

Weaknesses:
1. It is not clear how to extend this approach to decoder-only models which have become more prominent recently.


**Reproducibility:**

3: Could reproduce the results with some difficulty. The settings of parameters are underspecified or subjectively determined; the training/evaluation data are not widely available.

**Reviewer Confidence:**

3: Pretty sure, but there's a chance I missed something. Although I have a good feel for this area in general, I did not carefully check the paper's details, e.g., the math, experimental design, or novelty.

---

> ### Author Rebuttal · Authors · 2023-08-28
>
> We thank the reviewer for the insightful comments and appreciate your positive feedback. We would like to address the specific questions below.
>
> 1.Extend to decoder-only models
>
> Our approach can easily be extended to decoder-only models by replacing BERT with GPT. Representations can be obtained by getting the last position output of GPT. However, for a fair comparison, following all previous works, we use the BERT-base-uncased to obtain representations in this paper. Furthermore, GPT uses the decoder-only unidirectional Transformer model, which may limit its ability to capture global information and is mainly used for text generation tasks. On the other hand, BERT employs a bidirectional Transformer model, which allows it to capture contextual information from both left and right contexts. This enables BERT to better understand the semantic and obtain representations. Thus, we only consider using the BERT-base-uncased to obtain representations in this paper.

---

### Official Review · Reviewer_EoeX · 2023-08-05

**Soundness:** 3

**Excitement:**

3: Ambivalent: It has merits (e.g., it reports state-of-the-art results, the idea is nice), but there are key weaknesses (e.g., it describes incremental work), and it can significantly benefit from another round of revision. However, I won't object to accepting it if my co-reviewers champion it.

**Paper Topic And Main Contributions:**

This paper argues that, when using entity descriptions for entity linking, breaking down long descriptions into multiple key views is better than taking the entire description into a cross-encoder. The authors propose a key view selection algorithm, and show it brings up entity linking performance on a dataset with multiple domains.

**Reasons To Accept:**

- The challenge is real and important.
- The method is overall reasonable with good experimental support.

**Reasons To Reject:**

- Clustering sentences based on semantics or topics is not new. Selecting the most relevant sentences from a long document is also not new. I recommend the authors review relevant work and discuss how their key view selection method connects to and differs from previous work.
- The hard coding of V0 into the entity linking module looks unnatural. It works because the corpus consists of encyclopedias. Such hard coding can be questionable on other corpora. I would expect a good key view selection method should be able to figure out V0 automatically if V0 is indeed very useful.

**Reproducibility:**

3: Could reproduce the results with some difficulty. The settings of parameters are underspecified or subjectively determined; the training/evaluation data are not widely available.

**Reviewer Confidence:**

3: Pretty sure, but there's a chance I missed something. Although I have a good feel for this area in general, I did not carefully check the paper's details, e.g., the math, experimental design, or novelty.

---

> ### Author Rebuttal · Authors · 2023-08-28
>
> We thank the reviewer for the insightful comments and would like to address the specific questions below.
>
> 1.Key view selection method connects to and differs from previous works.
>
> Thanks for your suggestion. While we acknowledge that clustering sentences and selecting the most relevant ones are not new, we believe our approach is unique in the context of entity linking. To the best of our knowledge, this is the first paper to select mention-related information from entity description in this task. Additionally, our method differs from selecting the most relevant sentence methods by introducing the novel idea of forming sub views and selecting relevant views. Each view contains sentences that describe the same aspect. Utilizing all sentences in the selected views can help our model understand these aspects better to capture more fine-grained semantic information. We will further elaborate on this in the revised version of this paper.
>
> 2.Hard coding of V0
>
> Considering that V0 generally describes the entity, we manually add it to key views to make the final entity linking decision. As you described, it works may because the corpus consists of encyclopedias. However, for other corpora where V0 is not a summative description, there is no necessity to manually add V0 to key views. In such cases, the proposed key views selection method is able to figure out key views automatically.

---

### Official Review · Reviewer_7XKJ · 2023-08-07

**Soundness:** 4

**Excitement:**

4: Strong: This paper deepens the understanding of some phenomenon or lowers the barriers to an existing research direction.

**Missing References:**

Previous work introducing entity aspect linking:

- https://madoc.bib.uni-mannheim.de/44381/1/entity-aspect-linking.pdf
- https://www.dcs.gla.ac.uk/~shubham/publications/pdf/cikm2022.pdf

**Paper Topic And Main Contributions:**

This paper presents a methodology to include entity aspect (here called sub view) information for zero-shot entity linking.

**Reasons To Accept:**

- (+) A novel approach to include aspect entity in a transformer-based zero-shot EL framework.
- (+) Thorough evaluation and competitive results.

**Reasons To Reject:**

- (-) Single approach to capture aspect / sub-view information (since this is the heart of the paper, it would have been more informative to explore more/different solutions).

**Reproducibility:**

4: Could mostly reproduce the results, but there may be some variation because of sample variance or minor variations in their interpretation of the protocol or method.

**Reviewer Confidence:**

4: Quite sure. I tried to check the important points carefully. It's unlikely, though conceivable, that I missed something that should affect my ratings.

---

> ### Author Rebuttal · Authors · 2023-08-28
>
> We thank the reviewer for the insightful comments and sincerely appreciate your positive feedback. We would like to address the specific questions below.
>
> 1.Single approach to capture sub-view information
>
> In practice, we have explored many other solutions. However, due to the spatial limitations, we only present the most effective solutions in the manuscript. In our sub view clustering module, we tried different solutions to form sentence chunks. In our mention-aware key views selection module, we explored several heuristic methods. In our entity linking module, we explored multiple solutions at both the feature level and sentence level to aggregate key views. We will include a detailed description of these solutions in the appendix of our revised version to provide more information.
>
> 2.Missing references about entity aspect linking
>
> Our proposed method differs from entity aspect linking. In entity aspect linking setting, each entity has multiple predefined aspects. The goal of entity aspect linking is to link a mention to one of these aspects to obtain more fine-grained information. In contrast, our method aims to select mention-related views of each entity and aggregate them to determine whether the entity matches the mention without irrelevant information noise. We will describe the difference in the related work section of our revised version.

---

### Meta-Review · Area_Chair_axaA · 2023-09-17

**Recommendation:** 4

**Metareview:**

The paper under review presents a methodology to include entity aspect information for zero-shot entity linking. The proposed approach breaks down long entity descriptions into multiple key views.

In terms of soundness, overall, the paper is regarded as having a good level of support for its major claims. All three reviewers recognize that the paper contributes novel work to the topic of entity linking. The study has been praised as being well-structured, and having a relevant and important challenge at its core.

Despite the acknowledged strengths of the paper, the reviewers collectively pointed out certain weaknesses. Notably, one reviewer noted that the manuscript could benefit from reviewing relevant previous work in the area of selecting relevant sentences and clustering them semantically. Lastly, a concern was raised about how this approach would extend to autoregressive entity linking.

---

### Decision · Program_Chairs · 2023-10-07

**Decision:**

Accept-Findings

**Comment:**

The paper under review presents a methodology to include entity aspect information for zero-shot entity linking. The proposed approach breaks down long entity descriptions into multiple key views.

In terms of soundness, overall, the paper is regarded as having a good level of support for its major claims. All three reviewers recognize that the paper contributes novel work to the topic of entity linking. The study has been praised as being well-structured, and having a relevant and important challenge at its core.

Despite the acknowledged strengths of the paper, the reviewers collectively pointed out certain weaknesses. Notably, one reviewer noted that the manuscript could benefit from reviewing relevant previous work in the area of selecting relevant sentences and clustering them semantically. Lastly, a concern was raised about how this approach would extend to autoregressive entity linking.